# Effects of Social Media on Enotourism. Two Cases Study: Okanagan Valley (Canada) and Somontano (Spain)

**F. J. Cristófol** [1] , **Gorka Zamarreño Aramendia** [2,*] **and Jordi de-San-Eugenio-Vela** [3]

1   ESIC, Business & Marketing School, Market Research and Quantitative Methods Department, 28223 Pozuelo de Alarcón (Madrid), Spain; fjcristofol@esic.edu
2   Department of Theory and Economic History, University Malaga, 29013 Malaga, Spain
3   Communication Department, University of Vic; 08500 Vic, Spain; jordi.saneugenio@uvic.cat
*   Correspondence: gzama@uma.es; Tel.: +34-607-91-40-68

**Abstract:** The aim of this article is to analyze the social media effects on enotourism. Two territories of similar extension and with historical coincidences in their development have been selected: the Okanagan Valley, Canada, and the region of Somontano, Spain. Methodologically, an analysis of the content on Twitter has been performed, collecting 1377 tweets. The conclusion is that wineries create sentimental and experiential links with the users, avoiding commercial communications. Specifically, Okanagan wineries establish a relevant conversation network on Twitter based on the high percentage of responses, which is 31.3%, but this is not so in the case of Somontano, which is 12.8%. The tourist attractions most used to create a bond are the wine landscape and the gastronomy in the case of both territories. The tourism sustainability variable remains a minor matter in the emission of messages on Twitter.

**Keywords:** social network analysis; sustainable tourism; web 2.0; enotourism; Twitter; Somontano wines; Okanagan Valley wines; wines of British Columbia

---

## 1. Introduction

In the last decades, wine tourism has acquired great relevance within the international tourism panorama, becoming a segment of special interest [1], as it involves a series of activities that include visits to wineries, festivities related to wine and specifically wine-related events [2]. Enotourists have some perfectly defined requirements [3] and they want to experience the wine world in all its dimensions [4].

Wine tourism is a growing activity that has become an interesting way of development for the winemaking rural regions [5]. The relevance of wine in culture has exceeded the original framework of its farming to expand throughout the world, not just as a model of consumption, but also as a form of development for the territory. In fact, in recent decades, Europe, the USA, Canada or Latin America are pursuing new ways of rural development. This model needs to be based in sustainable and economically profitable activities. This new model of rural territory development should include policies, proceedings and impacts on the territory that reach those goals promoted by several administrations, since the policies implemented up to now have been insufficient. It is intended to change the productive and territorial structure in order to make the preservation of the environment and the patrimonial and cultural values the engine for economic development [6–9].

Tourists have a growing interest for products that correspond to certain criteria, such as culture conservation, lifestyle, local customs, environment and landscape [10,11]. Wine tourism mixes those

elements and offers a life experience that allows visitors to immerse themselves in activities where sensory, symbolic and hedonistic attributes become of maximum relevance [11–13].

Wine routes constitute a major opportunity to preserve traditions and to promote typical products, like local wines and gastronomy [14,15]. Wine and gastronomy have become, in many cases, an inseparable pair [16,17]. Therefore, they are tools to promote the kind of tourism that searches for an added value and that aims to become an important element in local development, with wine as the core idea.

The structure of this research is: firstly, a review of the literature, developed in this chapter; secondly, the presentation of the research methodology, as well as the explanation of the territories under study. The results are then presented, interpreted, and discussed, ending with the conclusions and implications of this paper.

## 2. Theoretical Background

Historically, we do not know precisely when the enotourist routes started in Spain, as wine has been a basic product in Mediterranean culture and circuits around quality wines are inherent to this area [18]. For over 25 years, ACEVIN (Spanish Association of Wine Cities) have developed the enotourism model in Spain. Their goal is to promote wine culture and tourism, becoming an example of collaboration between the public sector, the private sector and local citizenry [19–21]. As an object of academic interest, the first studies carried out in Spain were in areas widely known for their wines. Examples of such cases are La Rioja [22,23] and the Sherry Triangle [23]. In recent years, multiple studies about enotourism have been carried out along all our geography, like the studies in Somontano area [24–26].

In the case of Canada, the first traces of enotourism initiatives date from the final decades of the last century. They are specially concentrated in two geographic areas: the Niagara Peninsula in Ontario and the Okanagan Valley in British Columbia. The first few wineries were built in the area during the 1980s, but its peak started in the 1990s [27], experiencing a significant expansion since the first decade of the 21st century [28,29] that has made them a model of enotourism in North America [30,31]. The academic interest started at the dawn of the 21th century, with the seminal study of Telfer [28], a researcher who investigated the development of Canadian wine regions, specially the Niagara region [21,29,32,33], that has also been studied in depth [16,21,34–39] along with the Okanagan Valley [16,27,40,41].

We must understand that there are three different but complementary realities regarding wine. In the first place, wine as a gastronomic product, linked to wine routes [42], supported by the new tourist's consumptions habits and much more demanding in terms of quality, differentiation and personalization than before [12,43,44]. Secondly, wine as a cultural concept and a way to understand and get closer to the rural traditions of the area. A substantial proportion of enotourists choose a particular destination depending on the possibilities that it offers [40]. Therefore, enotourists are not just interested in learning about wine and its culture, but also in learning about the local culture of the region where the wine is produced [45,46]. Thirdly, wine as a hedonistic and epistemological experience [47] (9). Both needs are interrelated: enotourists prefer tourism that provides personal growth, in other words, they look for educational entertainment, 'edutainment' [48]. Given the multifaceted essence of wine, one of the keys to success is advertising [49,50] and experiential marketing [51,52].

The questions of this research are as follows:

1. How do wineries use social networks to attract tourists?

　　a. Which digital social networks are the most used?

2. Which tourist attractions do wineries emphasized in their audiovisual and graphic posts?
3. Is the concept of sustainable tourism included in the messages transmitted in social networks?

Since social networks are widely used, their impact and their role, has been studied in tourism [53] also in relation to wine tourism and the world of wine [54]. The impact on sales [55] or the perception of tourists [56] are recurring themes. The novelty of the research we have approached is to propose a comparative perspective study between two wine-growing areas, Somontano (Spain) and Okanagan Valley (Canada) that have consolidated the wine tourism segment. Although the use that the wineries make of digital marketing tools [57] and social media in particular [58] has partially studied, even in its relationship with Twitter [59], no analysis of the content of social media messages has been carried out comparing two areas that are involved in wine tourism interests.

## 2.1. The Hedonistic Experience of Wine

Enotourists are not only interested in the product, but in a mix of product, atmosphere and experience. Bruwer [47] emphasizes that, after wine consumption and purchase, tourists choose the vineyard they are going to visit based on the whole experience, which remarks the enotourists' interest in seeking hedonistic experiences. Researches demonstrate [60] that, once they make the decision of visiting the wineries, the wine quality falls into a second position and the services offered by the brand acquire greater relevance.

There is a debate about the prevalence of the different senses when shaping the wine experience. On the one hand, some analysis show that the sense of sight is the most relevant, followed by the sense of taste and the sense of hearing [61]; on the other hand, some consider the tasting and wine consumption the most important element of the wine experience [9,53]. However, cultural heritage remains attractive to the tourist, especially the US wine tourists are likely be sightseeing and visiting historical attractions while enjoying a wine holiday [62].

These studies are the clear evidence that enotourists' interest in living hedonistic experiences is also linked to living epistemological experiences. Enotourists are consumers seeking pleasure, enjoyment and personal development [45]. In order to satisfy the client, creating a positive experience that translates into an increased loyalty and relationship with the company [63], wineries need to consider the environmental framework that surrounds them. They must pay great attention to their facilities' design [64] and to the choice of a human team, so they must be careful to select caring professionals that know how to share their knowledge with great passion. That increases the chances of product purchase after the visit and, especially, the beginning of a strong client–winery relationship [51,65].

The wine tourist commonly makes their purchasing decisions a week before the visit [66, 67], as it is an impulsive consumption behavior that corresponds with the hedonistic desires and sensory impulses [68]. The enotourist, just as the compulsive shopper, wants to enjoy the shopping process—both guided by hedonistic values. Therefore, the most important thing is not the product acquired, but the pleasure of the experience [65]. This kind of purchaser is influenced by sensory impulses and is susceptible to being manipulated through marketing and communication techniques' stimuli [69,70].

## 2.2. The Promotion of Enotourism in Digital Environments

Enotourism is not just limited to wine consumption in a particular moment in time and location; it is a holistic product composed of three stages: pre-visit, visit and post-visit [16,71]. In addition, enotourists are not just visitors from one region or territory; they are wine consumers [72,73] that go to the winery seeking new experiences related to the product that provides them pleasure [65]. Enotourism does not focus just in visiting wineries or vineyards and tasting, it is a whole experience in sensorial levels, linked to the terroir. Therefore, the atmosphere that surrounds the region's traditions, gastronomy and culture needs to be emphasized [74], since it constitutes an essential element to build an appropriate communicative relationship between wineries. This also applies to the importance of satisfactory experiences in the consumer's behavior. These positive experiences increase the consumer's brand loyalty and inherently enhance the possibility of attracting new consumers, thanks to the positive

recommendations from satisfied customers [75]; in other words, there is a positive correlation between experience, brand loyalty and recommendations [76,77].

The use of technology can alternatively facilitate the development of enhanced experiences [78]. Internet and social networks has changed the customer's way of experiencing each of the stages that form the purchasing experience, changing the information search and product purchase method and the way of consuming the product, thereby recalling the whole experience [79]. In the pre-travel stage, destinations benefit when tourists search for information that inspires them about the place they are going to visit; during the journey, when they publish and share information; and returning home, when they share the experiences they have lived [80,81]. Nowadays, tourists have the chance to give their opinion and request information about any aspect of the destination, make suggestions or criticize, or even become opinion leaders [82,83]. The trip-planning process has been completely redefined [84]: the Internet has become the essential search tool and social networks are the place to share the experience lived [82,83].

Enotourists are often sociable [85] and active users of the Internet as their main information source in the pre-visit stage [86]. Word of mouth continues to be the main source of information to decide the consumption, so friends and relatives become the main product prescribers, especially in the pre-purchase stage [47,74].

The wine tourist is primarily influenced by comments and recommendations of third parties, compared to the purchase decision of other products [87]. In the case of complex products like enotourism, which is a mix of experiences, consumers are more risk reactive, so they are more likely to contrast the information before taking the decision. The search for comments in the networks helps reduce uncertainty when choosing the experience [88].

The channel where the comments are posted is relevant to the credibility perceived by the client [88]. There are two types of channels: channels sponsored by the brand itself, as advertising strategies in social networks, in the winery's account or with the help of influencers; and not sponsored channels, where there are real conversations about previous consumers' experiences [89]. Customers tend to trust more in unsponsored channels, which is why these comments have greater impact in the consumers' purchase decision [90]. Wineries should consider this factor when managing their presence in social networks. On the one hand, wineries should pay attention to the content that their customers can find on their digital channels and, on the other hand, they should promote attractive interaction that builds an active digital community to generate positive word of mouth (eWOM) about the brand [91]. Wineries need to have a full profile in not sponsored channels. To achieve that, they have to motivate clients to comment on these channels and facilitate its access by using links in the brand channels [92], a milestone to build a credible digital community with relevance in the customer's decisions.

There is a growing number of enotourists that only consider visiting wineries, going to events and actively participating in the brand's community if the brand publishes customer-driven content in social networks. This content must show the daily life of the winery, making it seem familiar, and include pictures taken by the clients to increase the degree of commitment and integration with the community [92].

Nowadays, tourists are more demanding, less patient and they give great value to their time and money. Therefore, they investigate and compare, looking for the most sophisticated experience that meets their needs, searching for the most personalized service possible [48]. Due to this trend, wineries that wish to obtain the best possible performance from the TICs need to develop all their brand and product design strategies focusing in the consumer (consumer centricity). In order to implement this strategy in the digital experience development, both for the company and for the customer, there are three criteria that must be fulfilled [93]:

- Building a powerful database to collect the clients' information in every stage of the experience. This will help companies understand the behavior and preferences of their customers by analyzing previous consumers and will allow them to adapt their services to the customers' demand, in order to make their offer as personalized as possible for each customer [94]. The combination of this

system with the database allows more flexibility when adapting to the consumer's culture in the center of the experience, which is the ultimate goal of any wine tourism company [48].

- Defining what kind of online presence they want. It is not enough to be present in the networks: presence needs to be managed and controlled both in private networks and in independent networks [90].
- Offering a fast answer and different forms of contact. Enotourists are demanding and value their time. Therefore, it is necessary to give them an answer as fast as possible. Despite the fact that there are systems that allow fast automatic answers, like chatbots, customers still prefer a personalized answer [95]. Furthermore, we must not forget that the core product of enotourism is wine, and its consumers are demanding when it comes to evaluate the staff's knowledge about the product [61]. Along with the telematic media, the winery must offer various options for the clients to submit any doubts or complaints, since they must feel free to choose how to communicate with the winery [95].

According to the previously mentioned differentiation criterion, on the one hand, we have the information of the company's website and, on the other hand, we must analyze the social network and the company independent platforms' content [96]. However, social networks have more influence than websites in their intention to visit [97]. Having a web domain is indispensable to almost every business [98] and its correct use gives a positive image and increases the intention to visit [97].

Wineries should complement the websites' information with the content of the social networks, which must be linked to the websites to make customers participate in both platforms [99]. Social networks have become the most influential phenomenon over the last few decades [100]. We must keep in mind that one of the most important features of social networks is the continuous interaction of the community members, the existence of formal and informal conventions, and an evident desire of the users to interact sharing experiences [101].

Another strategy that wineries should use to express the objective message and position themselves in the digital market is the use of "influential people marketing". Nowadays, the use of social media provides mass leaders (influencers), who are able to promote products in a more informal and direct way, alluding to their own experience. This kind of strategy has a great influence on the purchase decision [102]. The brand representative in networks, as well as the community managers, must be able to answer basic questions about the values and products of the winery [92].

In addition to social networks controlled by the company, where consumers are contributors, there are independent platforms where the customer is the one who initiate the conversation. Information and opinions on this kind of platforms have more credibility, as the customer does not relate them directly with a marketing or advertising strategy [90,103]. To any winery, it would be indispensable to achieve as much positive comments as possible in platforms like TripAdvisor, considered the world's largest tourist community [104]. Wineries must make an effort to spread, on their social network profiles, all the information related to the activities that they offer, in a way that is attractive and easy to interpret, relying on the use of images and videos. Wineries should also promote and encourage their customers' participation in order to increase the network communication [105,106].

Therefore, it is of paramount importance to keep an active participation on the media and social networks, obtaining a stronger web presence and increasing the destination's visibility. Enotourists, just like any kind of tourist, want to share actively their experiences, information, contents and news, using principally YouTube, Flickr, Facebook, Twitter or TripAdvisor [58,59,107,108].

## 3. Materials and Methods

This research is based on the analysis of the tweets published by the five most active wineries on Twitter from each of the territories studied between 1 June and 31 October 2019. The period chosen coincides with the grape harvest of both territories. For this purpose, a content analysis sheet adapted from Huertas Roig [109] has been used in order to standardize the analysis of the messages. The terms related to tourist attractions and emotional values that appear in the tweets have been taken into

account and, on the other hand, the type of written and audiovisual content have been classified differentiating the type of hypertext attached to the message.

This document focuses on the analysis of the messages transmitted via Twitter, since the value of this social network has been demonstrated in terms of its content and connectivity [59]. In addition, it is possible to know its creation of value by tracking and analyzing the activity of the publications. Wilson and Quinton [60] also claim that Twitter can enhance the creation of value for wine brands in terms of sharing feelings and perceptions, but not from a commercial point of view. According to Wimmer and Dominick [110] (1996:172): "Content analysis is a method of studying and analyzing communication in a systematic, objective and quantitative way, with the aim of measuring certain variables", so a systematization of the analysis has been made through the sheet in Table 1.

**Table 1.** Sheet of content analysis.

| Tourist Attractions | Emotional Values | Type of Written Content | Type of Audiovisual Content |
|---|---|---|---|
| Landscape and nature | Romanticism | Question | Picture |
| Wine landscape | Charm | Suggestion | Video |
| Intangible heritage | Friendship | Gratitude | Link |
| Popular culture | Escape | Information | |
| Cityscape | Modernity | Answer | |
| Urban leisure | Cosmopolitanism | Others | |
| Nightlife | Responsibility | | |
| Shopping | Authenticity | | |
| Sun and sand | Safety | | |
| Weather | Joy | | |
| Gastronomy | Youth | | |
| Ecology | Energy | | |
| Hospitality | Relax | | |
| Luxury | Tradition | | |
| Sport show | Quality of life | | |
| Sport practice | Sophistication | | |
| Plans | Diversity | | |
| Others | Others | | |

Source: Authors following Huertas Roig Model [109].

For this study, a sample of all the wineries from the Somontano Certificate of Origin has been collected, creating a digital tools directory for each of them: all 31 wineries have a website, but only 21 have their own Facebook page, 17 have a Twitter user and 14 have a presence on Instagram. In the case of Okanagan wineries, the sample is more complicated to collect because of the difference in sources. According to BC Wine Institute, there are 184 licensed vineyards and out of these, 154 have their own Facebook page, 143 on Twitter and 126 on Instagram.

Once the directory was made, the tweets of all the wineries with users on Twitter were downloaded with *Twlets*, a tool that allows the download of the last 3200 tweets from each account in a spreadsheet with the following fields:

- ID: this indicates the URL of the tweet.
- screen_name: this indicates the username of the account that posted the tweet.
- created_at: this indicates the date and time of the posting of the tweet.
- fav: this indicates the number of times it was marked as a favorite.
- rt: this indicates the number of times it was retweeted.
- RTed: this indicates if the post is a retweet from another account and, if so, indicates from which account.
- text: this shows the text of the post, including hashtags and URLs.

- media: it shows up to four columns, with the URLs of pictures and videos attached to the post.
- sentiment: it is divided into four columns where an algorithm classifies the messages, depending on if they are positive, negative or neutral and adds them a value named *Compound*.

There was, then, a sorting of the content, keeping only those posted between 1 June and 31 October 2019, coinciding with the high tourism season and the grape harvest in the different selected territories.

From the total analysis, five wineries in the Okanagan area and with the Somontano Certificate of Origin were chosen according to the number of messages posted, resulting in the following sample. The five most active wineries in Okanagan were Mooncurser, Fairview, Quail's Gate, Wildgoose and Ok Crush Pad, posting 794 messages in the period studied, with a total amount of 4297 tweets posted by all 143 wineries. In Somontano, the five most active wineries in that period were Viñas del Vero, Bodega Sommos, Bodega Pirineos, El Grillo y la Luna and Monte Odina, with a total amount of 583 tweets out of 591 messages from the 17 wineries with users on Twitter. Of all 1377 posts, 430 of them are retweets made by users of the wineries to other accounts. Finally, having eliminated the retweets, 947 posts corresponding to the 10 wineries studied were analyzed. Table 2 presents a summary of data for the accounts studied.

**Table 2.** Data Overview.

| Winery | Followers | Following | Total Tweets (until 04/07/2020) | Tweets between June 1 and October 31, 2019 |
|---|---|---|---|---|
| Viñas del Vero (S) | 8861 | 3637 | 9632 | 237 |
| Bodega Sommos (S) | 1618 | 774 | 1563 | 132 |
| Bodega Pirineos (S) | 3875 | 199 | 2792 | 131 |
| El Grillo y la Luna (S) | 650 | 252 | 687 | 55 |
| Monte Odina (S) | 259 | 341 | 381 | 28 |
| Mooncurser(O) | 3732 | 3272 | 5662 | 223 |
| Fairview Winery(O) | 3022 | 154 | 3708 | 158 |
| Quails Gate (O) | 12,200 | 1453 | 3188 | 150 |
| Wild Goose Wines (O) | 5019 | 3594 | 5072 | 138 |
| Ok Crush Pad (O) | 5102 | 3150 | 6055 | 125 [1] |

[1] Source: Authors.

*Territories under Study*

Okanagan Valley is located in British Columbia, a region in southwestern Canada, bordering the US. Vancouver is the most populated city of that state and its administrative capital is Victoria. Kelowna is the most populated city of the Okanagan Valley region, with more than 120,000 inhabitants. Along the Valley's 160 km from north to south, there are 3573 hectares of land planted with grape, according to Wines of British Columbia [111]. There are, on the other hand, a variable number of wineries that, according to the Taste Advisor platform, "should be between 175 and 195" because there are constant changes, such as wineries that are sold, others that are getting started and have not registered yet, and others that are disappearing. In this respect, according to the Wine BC explorer, there are 184 licensed vineyards, a fact verified with the wineries localization reports of the Employment, Business and Economic Development Department of the British Columbia government [112].

Somontano de Barbastro is a region in the autonomous community of Aragon. Its capital is Barbastro, with 16,979 inhabitants, and the closest province capital is Huesca, with 52,463 inhabitants. The region is located in the northwest of Spain, on the border with France, separated naturally by the Pyrenees. As opposed to the Okanagan Valley region, the wine region of Somontano is controlled by the Protected Designation of Origin Regulatory Council of Somontano (Consejo Regulador de la Denominación de Origen Protegida Somontano), formed by 31 wineries that occupy 3915 hectares, according to the annual report of the Council itself [113].

The selection of these two areas is initially based on the vineyards plantation extension and, due to their similarity, the work is focused in the specificities of each, aiming to compare their use of the different tools studied in this document. Even if it has a tradition of over 2500 years, it was not until "the 19th century, after the phylloxera attack in the Aragonese vineyard, when there was an event that would be the key to the wine sector of Somontano. The family and winery Lalanne, from Bordeaux and Buenos Aires, arrives in Somontano and introduces new varieties of grape such as Chardonnay, Merlot or Cabernet-Sauvignon" [114].

In the case of the Okanagan Valley, it is around the middle of the 19th century, in 1859, when father Charles Pandosy plants vines, in the mission that the French religious of the Congregation of the Missionary Oblates of Mary Immaculate had then in Kelowna.

## 4. Results

Below are the graphs that summarize the collected results. In this case, for an easier interpretation, the figures reflect the values that appear more than ten times in the sum of both territories.

Figure 1 shows the tourist attractions that are transmitted, 56.1% of them being related to gastronomy, with these posts linked to visits to wineries, tasting sessions and experiences related to food. In addition, 18.6% of the tweets are related to wine landscape and in the case of plans, such as events in wineries; the total is 9.7%, where vineyards and wineries either are mentioned in the text or present in the audiovisual material of the posts. In the case of urban leisure, Okanagan Valley does not mention this category, but Somontano does and relates to tapas routes and wine consumption in bars. In the category of popular culture, local or national festivals are mentioned: in the case of Somontano, the festivals of San Ramón or the Virgen del Pilar; in the case of Okanagan, Halloween, Thanksgiving or Canada Day. Ecology appears in 2.2% of the posts.

Figure 2 shows the emotional values in each tweet. Of the total amount, 58% of posts transmit quality of life, which is the case of those experiences related to gastronomy as a hedonistic experience. Tradition appears in 17.7% of the posts and it relates to concepts such as grape harvest or the age of the vineyards. Other values that appear are authenticity, in 5.7% of them, or dynamism, in 5.6%. In the case of friendship, which is present in 3.9% of the posts, it is reflected explicitly through words like friend or with pictures attached to the tweets analyzed. Modernity also appears in 2.9% of the tweets, and cosmopolitanism, in 1.9%.

Figure 3 classifies the tweets posted depending on the type of written content. According to this, wineries use the channel mainly as a means of information; those of Somontano focus 61.4% of their posts on this matter, while those of Okanagan do so in 21.4%. Okanagan wineries have a conversation network on Twitter, as 31.3% of their tweets are responses, compared to 12.8% in the case of Somontano wineries. 15.6% of Somontano's tweets are thank you messages, while in Canadian wineries' these are 20.14% of all their posts. Comments also account for 8.18% of Somontano wineries' tweets and 24.1% of Okanagan's. Questions from wineries make an insignificant number.

As shown in Figure 4, 49.8% out of 391 posts of Somontano present attached pictures, while in the case of Okanagan wineries, there are pictures in 36.8% of theirs. As far as posts with links are concerned, there are 17.6% from Somontano's accounts and 46.5% from Okanagan's. The percentage of posts without any associated hypertext (picture, video or link) is 28% for Somontano and 15.9% for Okanagan, as shown in Figure 5. Finally, videos appear only in 4.6% of all posts of Somontano and barely in 0.8% of Okanagan.

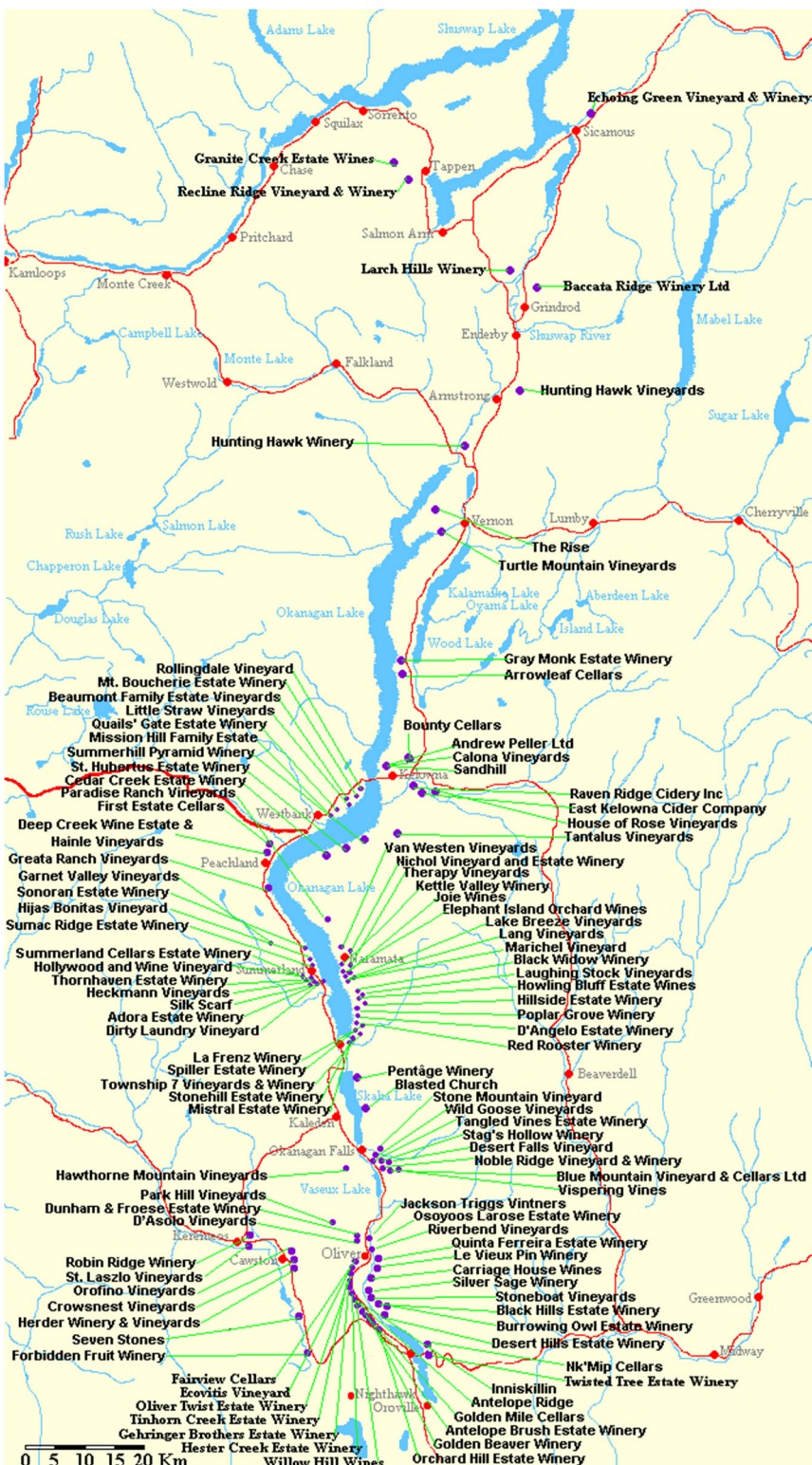

**Figure 1.** Okanagan Wineries map.

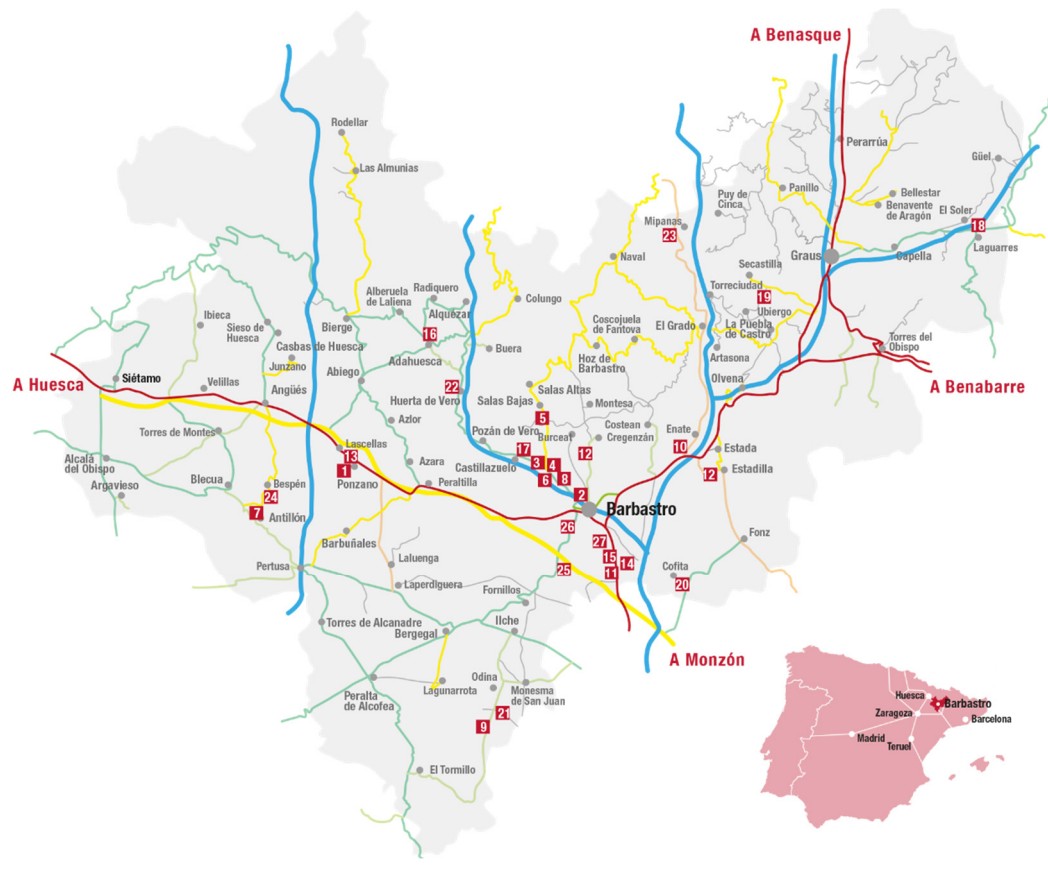

**Figure 2.** Somontano Wineries map.

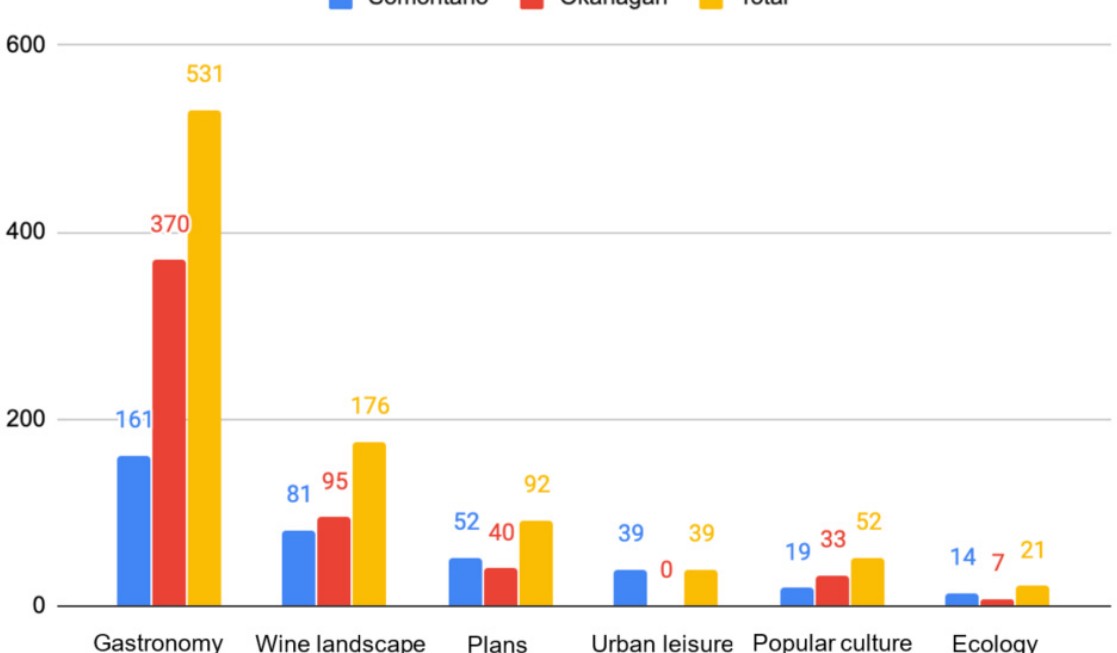

**Figure 3.** Classification of the tweets by tourist attractions.

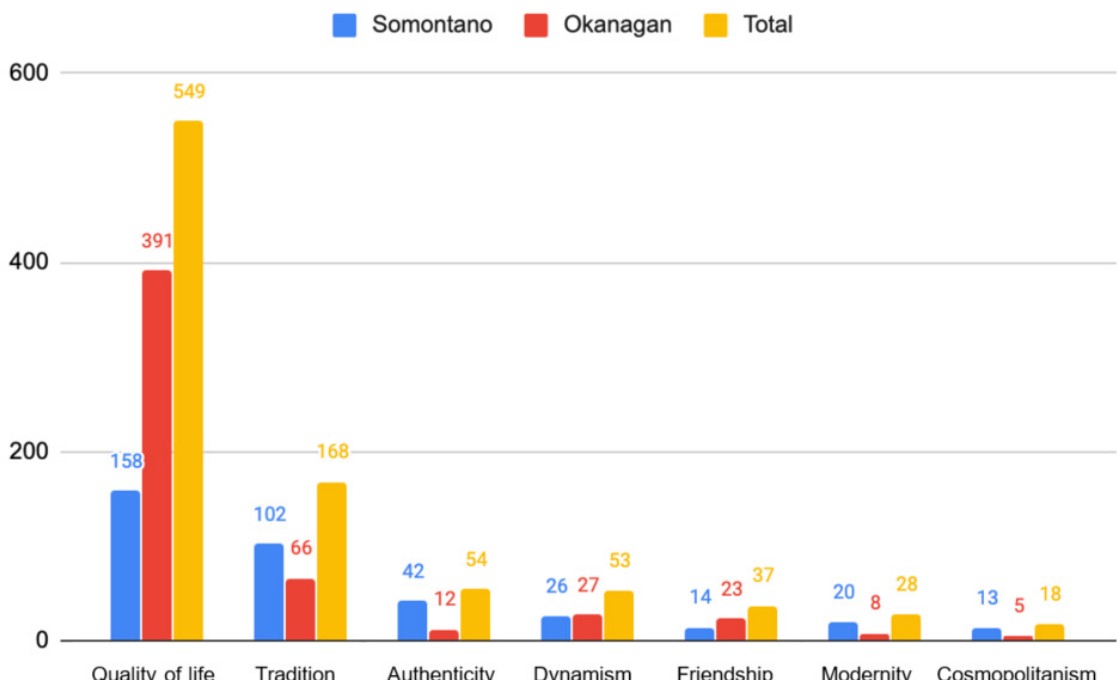

**Figure 4.** Classification of the tweets by emotional values.

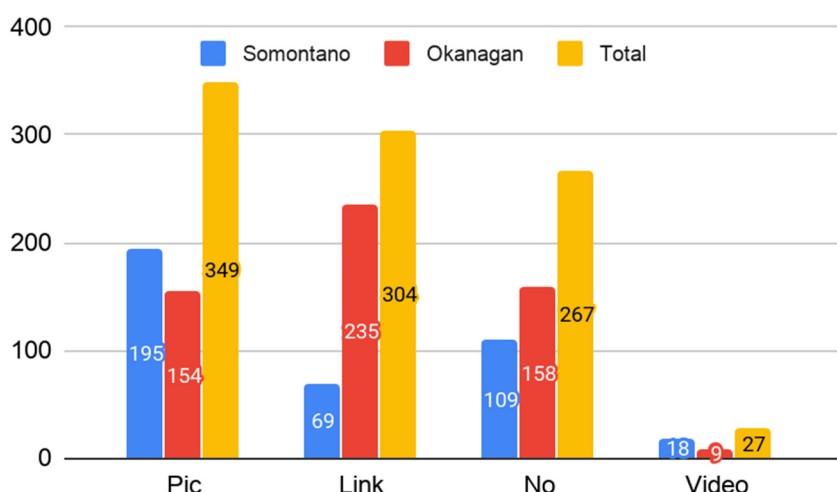

**Figure 5.** Classification of tweets by audiovisual content.

As a final analysis of the results, as far as Okanagan is concerned, it is striking that the Wild Goose winery mostly uses Twitter as a platform to repost their Instagram posts, so the original content for Twitter is minimal. In the case of Fairview Cellars, the user is addressed in the first person and, according the information shared in their biography, it is a personal account managed by Bill Eggert, "owner of this small 2500 case winery on the Golden Mile in Oliver, BC, Canada, producing blends and varietals from the 5 classic Bordeaux cultivars". For its part, Ok Crush Pad uses Twitter mainly as a conversational channel. In the cases of Quail's Gate and Mooncurser, a use as a corporate communications channel can be found. In the case of Somontano, the five wineries use their Twitter profile as an only corporate communications channel, without Instagram reposting or first-person comments, as shown in Figure 6.

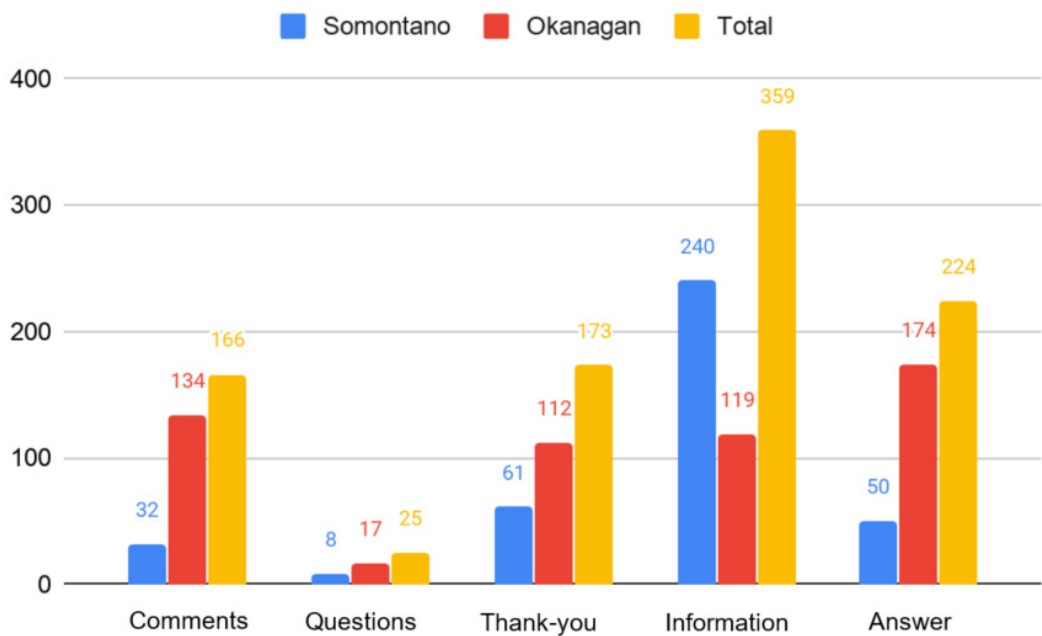

**Figure 6.** Classification of the tweets by its type of content.

## 5. Discussion

Enotourism, as the literature has pointed out, is a phenomenon that goes beyond the traditional tourist field and must become an activity with a positive impact on the economic and social development of the rural areas linked to wine production. In the case under analysis, this process of territorial enhancement is a market value resulting from the traditional tourism logic, which means that destinations are still being shaped by emphasizing the quantitative aspects and the strictly economic impacts generated by this industry. However, developing a sustainable tourism model that influences the territory, generating opportunities among the local population and establishing the appropriate synergies between visitors and the local population [115] requires the application of a transversal logic resulting from strategic planning and management of the territory. This is the way to value heritage, cultural and social values so that it generates a distinguishing identity and a proposal of value in itself.

As aforementioned, social networks have transformed the communication process in recent decades: relations between users are now direct, interactive and seek to generate differentiating experiences. Wine tourism has not remained oblivious to exploiting the possibilities offered by social networks. Therefore, wineries use their social network profiles to offer information about their activities. At the same time, they seek to promote and encourage the participation of their customers in order to increase the conversation on the network and the number of positive comments on other networks. However, enotourists are looking for a distinguishing experience that activates the senses and require a hedonic experience. To achieve this, it is necessary to build a homogeneous story and transmit it through social networks using all the potential of multimedia tools.

The story must coincide with the objectives of the strategic planning of the territory, combining the interests of wineries and wine tourists. For this purpose, it is necessary to provide content to the speeches broadcast in the different communication channels, especially in the social networks. In the study of the case, wineries emit stereotyped messages that do not meet the objectives that should govern the integrated communications, which combined the sustainable values of the terroir, transmitting a complete experience to the audience composed mainly of wine tourists.

Previous research [116–118] suggests that the social media universe is only used as an element of information and promotion of local services. In no case is the positioning or brand content worked on. Similarly, it denotes a limited or no interaction of the profiles analyzed on Twitter with their respective audiences, thus limiting the projection of the territorial brand value for the cases analyzed in this paper.

## 6. Conclusions

Wineries use Twitter to create a sentimental and experience-based link with their followers. Although it is not the most used social network, it does focus especially on creating engagement and not specifically on a commercial approach. Twitter is, as mentioned above, a social network that facilitates engagement and soft value. "The content of Twitter messages across all the different Twitter profiles frequently contained sentiment" [59]. Even though the most used social network is Facebook, used by 81.4% of the wineries (67.74% in Somontano and 83.7% in Okanagan), Twitter stands in second place, with 74.4% (54.8% in Somontano and 77.7% in Okanagan). Finally, 65.1% of the wineries use Instagram (45.2% in Somontano and 68.5% in Okanagan). There is a relevant difference in the use of social networks between one territory and the other, as well as in the number of posts of the five most active wineries, which in the case of Somontano ascend to 391 (plus 192 retweets) and in the case of Okanagan, to 557 (plus 238 retweets).

As far as tourist attractions are concerned, two main elements stand out: gastronomy and wine landscape. In the case of gastronomy, there is an evident alignment between the posted messages and the hedonistic or pleasant experience of consuming wine. Secondly, wine landscape is presented as a catalyst for the experiences lived and the territorial imaginary. Moreover, this tourist attraction appears in most cases linked to tradition as an emotional value. In the tweets posted to show the values of each of the wine territories we find that, in the case of Okanagan Valley, the environment and the landscape of wine are part of the experience of the winery itself, thus integrating itself into the territory. On the other hand, in the case of Somontano, it more usually refers to urban leisure and it moves the experience out of the winery to place it in a separate space, so that the natural environment becomes irrelevant, as shown in Figure 7.

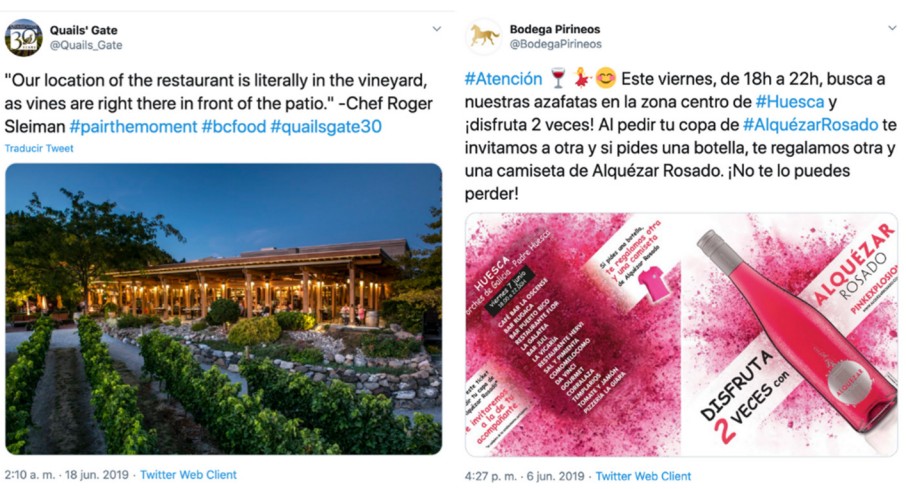

**Figure 7.** Posts comparison: Quail's Gate [118] (Okanagan) and Bodega Pirineos [118] (Somontano).

In none of the cases does sustainable tourism appear as a central argument. References do exist, but in no case are they relevant, so everything related to ecology, sustainability and respect for the environment drops to second place. However, in Okanagan's case, the value of ecology is implicit in the shared pictures and the shown landscapes, which is not the case with Somontano wineries, hardly presenting this type of value nor explicitly or implicitly. Therefore, this concludes that in the Canadian case there is a desire for the holistic consumption of wine, which is expressed with a continuous reference to the territory and a clear focus on the natural environment as a value of the product itself. Thus, Somontano generates a competitive and comparative advantage with their posts on Twitter, which appeal mainly to the marketing of wine, regardless of the territory to which it is associated.

With regard to the relational communication established on Twitter by the two territories, we must highlight the contribution of the Okanagan wineries, whose responses account for around a third

of their total posts and this implies a desire to establish a direct link with their audiences. On the contrary, in the case of Somontano, only one 1 of 10 posts on Twitter involves direct interaction with its stakeholders.

Based on the analysis of all the publications, a lack of a preconceived strategy in the line of transmitting a story that links wine and territory is noted. There is no explicit relationship between commercial brands and territorial brands, although in the case of the wines from Okanagan Valley, their link to the territory and the landscape is recurrent in tweets, something that does not occur in the case of Somontano, where the audiovisual material does not show a specific territory. In conclusion, in the Canadian case, there is a desire to show the link between the territorial identity and the wine, while this relationship does not exist in the Spanish case.

The opportunities arising from a territorial brand go beyond a business element. The vineyards studied should understand that the marketing element—exclusively purchase/sales—must be overcome in order to emphasize the intrinsic value of the terroir. The basis of the discourse must be built on non-consumable and identity elements, initially intangible. This can have a tangible return from the notoriety and impact that a well-recognized identity has based on a winemaking and wine tourism landscape, such as the one shown in this research. One of the main axes of advance in the management of the discourse of the organizations in Twitter is to understand this social network as an element of diffusion of place branded content, instead of being a simple platform of exclusively commercial messages and without any connection between them (absence of storytelling). Thus, the user is presented as an absolutely passive element, far from the reality of digital communication strategies that place the consumer at the center of them. It can also be observed that the stakeholders at no time enter into conversation with the Twitter profiles analyzed, thus perceiving the messages issued as something alien to their own reality and interest. Secondly, the possibilities that a territorial brand offers beyond commercial objectives are being wasted, ignoring other functionalities and objectives that can be assumed, such as those related to territorial governance, strategic territorial planning or citizen participation processes, among others.

**Author Contributions:** Conceptualization, G.Z.A.; Data curation, F.J.C. and J.d.-S.-E.-V.; Formal analysis, F.J.C. and J.d.-S.-E.-V.; Investigation, F.J.C., G.Z.A. and J.d.-S.-E.-V.; Methodology, F.J.C.; Project administration, G.Z.A.; Resources, G.Z.A.; Software, F.J.C.; Validation, J.d.-S.-E.-V.; Writing—original draft, G.Z.A.; Writing—review & editing, J.d.-S.-E.-V. All authors have read and agreed to the published version of the manuscript.

**Funding:** This research received no external funding.

**Conflicts of Interest:** The authors declare no conflict of interest.

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
