# Peer review of "Effects of Social Media on Enotourism. Two Cases Study: Okanagan Valley (Canada) and Somontano (Spain)"

_sustainability, doi:10.3390/su12176705_

Round 1

Reviewer 1 Report

In the introduction, at the end, I suggest to include the structure of the paper.

After the introduction, I recommend entering the theoretical background: ("The hedonistic experience of wine", "The promotion of enotourism in digital environment").

For the theoretical background see also:

Garibaldi, R., Stone, M. J., Wolf, E., & Pozzi, A. (2017). Wine travel in the United States: A profile of wine travellers and wine tours. Tourism management perspectives23, 53-57

Garibaldi, R., & Sfodera, F. (2020). Technologies for enhancing wine tourism experience. The Routledge Handbook of Tourism Experience Management and Marketing, 16.

Garibaldi, R., & Sfodera, F. (2020). tourism experience. The Routledge Handbook of Tourism Experience Management and Marketing, 409.

I suggest to insert the paragraph  "Territories under study" in the "Materials and methods".

It is essential to strengthen the discussion, inserting the links between theoretical and practical part.
In the conclusions, I suggest to better emphasize the theoretical and practical implications of the paper.

Author Response

Dear reviewer,

Thank you for the time you have taken to make the suggestions on our paper. We consider your comments very appropriate.

As indicated, we have included the structure of the paper in the place that it suggested. We have also found the references that you indicated to us of great interest, especially the first two that have been adequately cited in text.

We have introduced the section on the methodology as you indicated. In the same way, we have improved the discussion and the conclusions by collecting your suggestions trying to get sronger the theorical and practical implications of the paper.

Reviewer 2 Report

The paper "Effects of Social Media on Enotourism. Two cases 2 study: Okanagan Valley (Canada) and Somontano 3 (Spain)" regards the use of social media on the enotourism on 2 destinations in Canada and Spain.

The paper is well written and reports correctly the hypothesis of the work and the results given the data. 

By me the paper could be published after minore requirements:

  • pag. 4, row 122 the reference 58 is double.
  • check the reference list using the style as presented in https://www.mdpi.com/authors/references

Author Response

Dear reviewer, thank you for the time you have taken to make the suggestions on our paper. We consider your comments very appropriate.

As indicated, we have corrected the error in the bibliography due to a problem with the Mendey program and we have made a review of the rest of the references.   The suggestions of the rest of the reviewers have also been incorporated, as you can see in the revised manuscript.

Reviewer 3 Report

Thanks a lot for your interesting paper! However, there are some further issues which could improve the quality and readability of your paper and highlight related problems to a better extent, respectively.                      The content of the introduction is very complete and adequate. But, the paper needs to address more clearly the novelty of the research. In addition, in the introduction section, I recomend to insert the structure of the paper.    The discussion do not provide enough detail. You need a deeper level of analysis of the key issues, and a more balanced account of the literature that takes us beyond the description of this reference said this and that reference said that. Focus on the meaning that those references, collectively, allow you make of the arguments that you are developing.                          In the conclusion, I recommend to add reference for explaining theoretical implication, but also to consider an unique paragraph, including practical implication and limitations.                                                                    While I suggest these relatively minor revisions of the article, I remain excited about the manuscript’s contribution to sustainability. I hope the authors receive this review and strengthen the manuscript, and I look forward to seeing it in publication.

Author Response

Dear reviewer,

Thank you for the time you have taken to make the suggestions on our paper. We consider your comments very appropriate. As indicated, we tryed to  to address more clearly the novelty of the research. We included the structure of the paper also.

We have improved the discussion trying to provide enough details for a deeper analysis.

In the same way, we have improved d the conclusions by collecting your suggestions getting stronger the theorical and practical implications of the paper.

Finally, we would like to thank you for your words about our research as it encourages us to improve and continue working harder on our future proposals.

Sincerelly: The authors.

Round 2

Reviewer 1 Report

In the introduction, at the end, I suggest to include the structure of the paper.

After the introduction, I recommend entering another paragraph named "Theoretical background" and that includes "The hedonistic experience of wine" and "The promotion of enotourism in digital environment".

Author Response

Dear reviewer.

Thank you for your contribution to the structure of our paper.

We have made the pertinent changes so that the first section is now reorganized as you have suggested. We believe that this change brings a greater understanding of the structure of the paper.

Regards, 

The authors
